# Space-charge-limited electron and hole currents in hybrid organic-inorganic perovskites

Mohammad Sajedi Alvar[1✉], Paul W. M. Blom [1✉] & Gert-Jan A. H. Wetzelaer [1✉]

Hybrid organic-inorganic perovskites are promising materials for the application in solar cells and light-emitting diodes. However, the basic current-voltage behavior for electrons and holes is still poorly understood in these semiconductors due to their mixed electronic-ionic character. Here, we present the analysis of space-charge-limited electron and hole currents in the archetypical perovskite methyl ammonium lead iodide ($MAPbI_3$). We demonstrate that the frequency dependence of the permittivity plays a crucial role in the analysis of space-charge-limited currents and their dependence on voltage scan rate and temperature. Using a mixed electronic-ionic device model based on experimentally determined parameters, the current-voltage characteristics of single-carrier devices are accurately reproduced. Our results reveal that in our solution processed $MAPbI_3$ thin films transport of electrons dominates over holes. Furthermore, we show that the direction of the hysteresis in the current-voltage characteristics provides a fingerprint for the sign of the dominant moving ionic species.

[1] Max Planck Institute for Polymer Research, Ackermannweg 10, 55128 Mainz, Germany. ✉email: sajedi@mpip-mainz.mpg.de; blom@mpip-mainz.mpg.de; wetzelaer@mpip-mainz.mpg.de

Hybrid organic–inorganic perovskites have emerged as promising semiconducting materials over recent years[1,2]. Especially in optoelectronic devices, such as solar cells[1,2], light-emitting diodes[3,4], lasers[5,6], and photodetectors[7–9], perovskites show great potential. However, understanding the device physics of perovskite-based devices has proven not to be straightforward, which can mainly be traced back to the mixed ionic–electronic conduction behavior of these materials[10–18]. One particularly important aspect in understanding the behavior of perovskite-based electronic devices is the characterization of the transport of electronic charges. Over recent years, many techniques have been employed to measure the charge-carrier mobility in perovskites[19–30], giving a vast range of different numbers[31,32]. The large differences observed in the measured charge-carrier mobility may, in part, be the result of different perovskite formulations and processing conditions, resulting in different film morphologies. However, also experimental techniques to measure the mobility can give rise to different values[31,32]. For instance, time-resolved techniques may only probe fast charge carriers within crystal grains, whereas the slower transport of charges across grain boundaries and those affected by defect sites may be disregarded in the measurements. Such slower charge carriers may considerably impact the device performance.

A powerful technique that has proven its value in determining the time-averaged steady-state mobility of organic semiconductors is the measurement of space-charge-limited currents (SCLCs)[33,34]. SCLCs are observed in so-called electron- and hole-only devices, in which, by careful choice of the electrodes, either only electrons or holes are injected into the semiconductor. The maximum electrostatically allowed current in such a device is limited by the buildup of space charge. The space-charge density depends on the permittivity of the semiconductor, similar to a parallel-plate capacitor. The current will then be determined by the conductivity, being a product of the space-charge density and the charge-carrier mobility. In perovskites, classical SCLC theory has been used previously to estimate the density of defects, or trapping sites, from measured current–voltage characteristics[19]. An important feature in the SCLC model is that the electric field resulting from the injected space charge exhibits a square-root dependence on distance from the injecting contact. However, it is well established that ion movement plays an important role in the shape, magnitude, and hysteresis of the current–voltage characteristics of perovskite solar cells[14,17,35–37]. As slow-moving ions modify the electric-field distribution in the device as a function of

time, it is evident that classical SCLC models are not applicable to mixed ionic–electronic semiconductors, such as perovskites. The ion dynamics and resulting field distributions will strongly affect the current–voltage characteristics in single-carrier devices, greatly complicating their analysis. Application of the SCLC model disregarding the effects of ions on permittivity and field distribution then leads to erroneous results regarding charge-carrier mobility and trapping sites. Although there are numerous publications on SCLC measurements of perovskite single crystals and thin films, the effect of mobile ions, permittivity, and temperature on the current–voltage behavior are missing[7,19,38–54]. To date, an accurate description of SCLCs in perovskite thin films does not exist. In this study, we investigate SCLCs in single-carrier devices of methylammonium lead iodide (MAPbI$_3$), the work-horse material in perovskite solar-cell research. As a central result, it is demonstrated that the frequency and temperature dependence of the apparent dielectric constant is of paramount importance in understanding the magnitude, the scan-rate dependence, and temperature dependence of SCLCs in perovskites. We develop a drift-diffusion model, including experimentally validated ion dynamics that can consistently reproduce the scan-rate dependence and temperature dependence of the current–voltage characteristics of electron-only and hole-only devices. The quantitative agreement allows for reliable determination of the electron and hole mobility from SCLC measurements on MAPbI$_3$ thin films. It is also demonstrated that the direction of the hysteresis in the electron and hole currents reveals the sign of the dominant mobile ionic species.

## Results

**Analysis of the relative permittivity.** In the analysis of the current–voltage characteristics of MAPbI$_3$ perovskite solar cells in literature thus far, the dielectric constant ($\varepsilon$) was assumed to be frequency independent, having constant values typically ranging from 6 to 117[7,19,35,37–44,46,48–51,53–59]. Also for other perovskites, the relative permittivity in the SCLC analysis was taken to be constant[7,19,38–40,42–44,46,48–51,53,54,58,59]. In Fig. 1, the frequency dependence of the dielectric constant is displayed, as obtained by impedance spectroscopy on MAPbI$_3$ capacitors. The relative permittivity is observed to be rather constant for frequencies above 100 Hz, having a value of 65, much higher than that used in most device-modeling studies but consistent with earlier results[60]. However, at lower frequencies, the dielectric constant increases

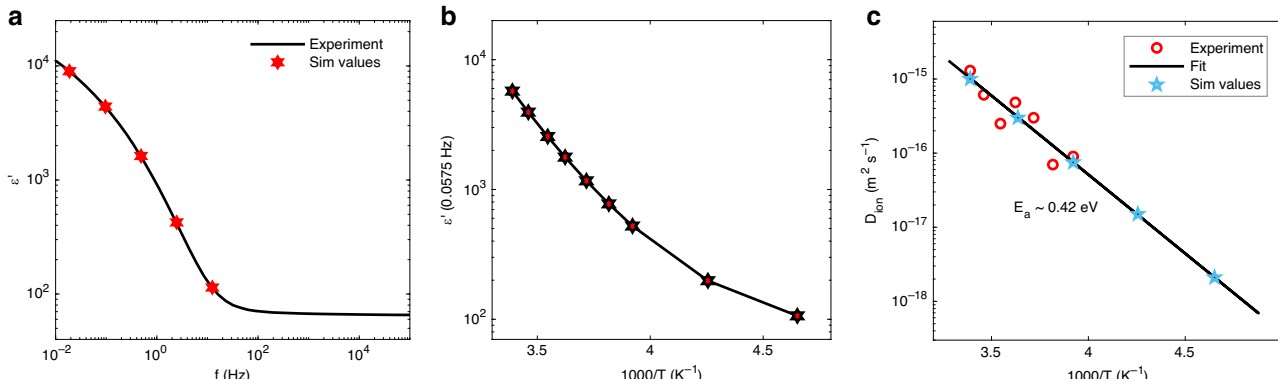

**Fig. 1 The apparent dielectric constant of a MAPbI$_3$ thin film. a** As a function of frequency, obtained by impedance spectroscopy (black solid line). The red stars correspond to the apparent dielectric constant at the voltage scan rates used in the simulations (Sim values) in Fig. 3. **b** Temperature dependence of the apparent dielectric constant (red filled stars) at 0.0575 Hz (dashed line in Supplementary Fig. 1), corresponding to the scan rate of the temperature-dependent *J–V* measurements in Figs. 2 and 4. **c** Ion diffusion coefficient as a function of temperature, obtained from impedance spectroscopy (red circles). The measurements are fitted with an Arrhenius law with an activation energy of 0.42 eV (black line). The blue stars represent the ion diffusion coefficients used in the simulations.

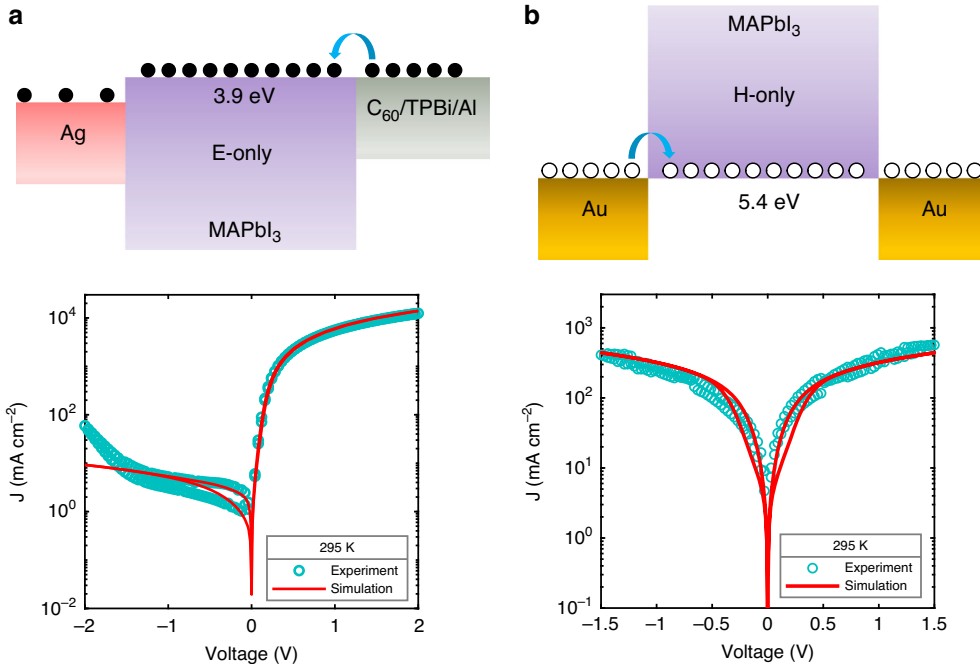

**Fig. 2 Schematic device structure and current density–voltage characteristics. a** Electron-only device, using a $C_{60}$(5 nm)/TPBi(5 nm)/Al electron-injection layer. Due to the barrier at the Ag bottom electrode, the current in reverse bias is injection limited. The dark cyan symbols represent the electron current measured at room temperature, with a scan rate of 0.46 V s$^{-1}$. The red solid line shows the simulated electron current using the electronic-ionic drift-diffusion model with an apparent dielectric constant of 5700. **b** Schematic of the hole-only device, using Au bottom and top electrodes. The symbols and solid line represent the measured and simulated hole current, respectively, under the same conditions as the electron current (**a**).

considerably, up to two orders of magnitude higher. This apparent high dielectric constant at low frequencies is the result of slow-moving ions[18]. As stated above, the dielectric constant has a direct influence on the magnitude of the SCLC, as it determines the amount of space charge that builds up in the semiconductor layer.

**Characterization of electron- and hole-only devices.** To measure space-charge-limited electron and hole currents, we have fabricated electron- and hole-only devices, as displayed in Fig. 2. To optimize electron injection, a thin (5 nm) $C_{60}$ layer was used, capped with a 5 nm 1,3,5-tris(1-phenyl-1H-benzimidazol-2-yl) benzene (TPBi) buffer layer[61,62]. The corresponding current density–voltage characteristics are displayed in Fig. 2c, d. The electron-only device shows asymmetric current–voltage characteristics, which is due to the electron-injection barrier at the Ag bottom electrode (see also Supplementary Fig. 4), resulting in a lower, injection-limited current in reverse bias. The hole-only device displays almost symmetric $J$–$V$ characteristics. The hole current is observed to be lower than the electron current. We note that we observed this behavior for many similarly prepared samples over the course of several years.

When measuring a SCLC in a single-carrier device, the scan rate is typically low (0.1 V s$^{-1}$ to 1 V s$^{-1}$). As a result, slow-moving ions can follow the changes in applied voltage, leading to small hysteresis in the measured current–voltage characteristics. This is shown for electron- and hole-only devices of MAPbI$_3$ in Fig. 2. Considering the slow scan rate of 0.46 V s$^{-1}$, the question arises which value should be taken for the dielectric constant in the SCLC analysis. Another important factor in the analysis is the influence of the moving ions on the electric-field distribution in the device, and thus the current. In many device-modeling studies, the ion diffusivity and concentration are not known, and literature values are taken, compromising the analysis. The situation is even more complicated when modeling complete

solar cells including charge-transport layers, as the electron and hole mobility of all materials, the recombination rate, and possible charge-trapping effects are also not known accurately. By using all of these quantities as fit parameters, a reasonable agreement with experiment may be obtained, although a reliable analysis is near impossible without experimentally validating the input parameters in the model. Therefore, we have recently determined the ion diffusion coefficient ($1 \times 10^{-15}$ m$^2$ s$^{-1}$; Fig. 1) and ion concentration ($2 \times 10^{25}$ m$^{-3}$) in similarly prepared MAPbI$_3$ films, verified by using two independent techniques[18]. These techniques involved the analysis of impedance spectroscopy with a simple equivalent circuit for ionic conductors and measurement of the electric displacement as a function of frequency, which could be reproduced with a mixed ionic–electronic drift-diffusion model. It was demonstrated that the movement of ions completely dominates the displacement characteristics. Furthermore, it was observed that the drift-diffusion model could only reproduce the displacement characteristics when a frequency-dependent permittivity was used, as displayed in Fig. 1a. In the analysis, it was assumed that positive ions are mobile and negative ions are fixed[18], based on previous studies. Likely candidates for the mobile positive ions are methylammonium ions or iodine vacancies[15,37,63].

As we have characterized the dielectric constant, ion diffusivity, and ion concentration experimentally, we can now fit the current–voltage characteristics of our single-carrier devices by tuning only the charge-carrier mobility. In this case, we have used an apparent relative dielectric constant of 5700, as measured at a frequency of 0.0575 Hz. This frequency corresponds to the voltage scan rate of the $J$–$V$ measurements, $f = \frac{\text{Scan rate}}{4 \times V_m}$, where $V_m$ is the amplitude of the applied voltage (see Supplementary Fig. 2). We obtain a mobility of $1.2 \times 10^{-6}$ m$^2$ V$^{-1}$ s$^{-1}$ for electrons and $3.5 \times 10^{-10}$ m$^2$ V$^{-1}$ s$^{-1}$ for holes. We note that our extracted mobilities are comparable to values obtained by electrode-based mobility measurement techniques[31,40,43,64,65] and lower than the

ones obtained by electrode-free techniques[27,28,31]. We note that the large variation in reported mobilities is likely the result of differences in sample morphologies, measurement techniques (e.g., different timescales), and their interpretation. For instance, as demonstrated here, classical SCLC analysis is not applicable to semiconductors with mobile ions. We further note that similarly low hole mobilities have been reported in MAPbI$_3$ field-effect transistors, also being a steady-state technique[66,67]. As demonstrated in Supplementary Fig. 7, the solar-cell characteristics are excellently reproduced by the drift-diffusion model, using the above-determined mobilities. More extensive solar-cell simulations are the topic of a future study.

Although our electron and hole mobilities are in a range similar to earlier reported values for MAPbI$_3$ thin films of ~$4\times 10^{-8}$ m$^2$V$^{-1}$s$^{-1}$ obtained from time-of-flight measurements[64], we do not find ambipolar transport, but a clearly higher mobility for electrons as compared with holes. We verified that our hole currents are not limited by a contact barrier by comparing the hole injection from Au, poly(3,4-ethylenedioxythiophene) polystyrene sulfonate (PEDOT : PSS), and polytriarylamine (PTAA) into MAPbI$_3$ (Supplementary Fig. 5). We note that the MAPbI$_3$ morphology and grain size is independent of the bottom electrode used (Supplementary Fig. 3). Furthermore, with

the inclusion of an injection barrier in the device model the shape of the J–V characteristics cannot be reproduced. We note that we cannot fully exclude that the low hole mobility originates from the presence of shallow hole traps[68], which in the classical model would give rise to an SCLC in which the mobility is replaced by an "effective mobility" defined by the product $\mu\theta$. Here, $\mu$ is the mobility without shallow traps, which is reduced by a factor $\theta$, being the fraction of free carriers with regard to the total amount of injected carriers. The effective mobility $\mu\theta$ can thus be considerably lower than the free charge-carrier mobility $\mu$. In this case, the measured current has all the features of an SCLC, except that the obtained mobility represents an effective mobility. However, this does not change our conclusion that the charge transport is highly unbalanced and electron dominated. Furthermore, it should be noted that the high current in forward bias in the electron-only device is limited by the series resistance of the electrodes, despite our attempts to reduce this resistance as much as possible in the fabricated devices. Therefore, the determined electron mobility should be viewed as a lower limit.

As an SCLC is essentially determined by the product of the permittivity and the (effective) mobility, a question is whether a fit could also be obtained when assuming a lower value for the permittivity, as obtained at higher frequencies (Fig. 1). As shown

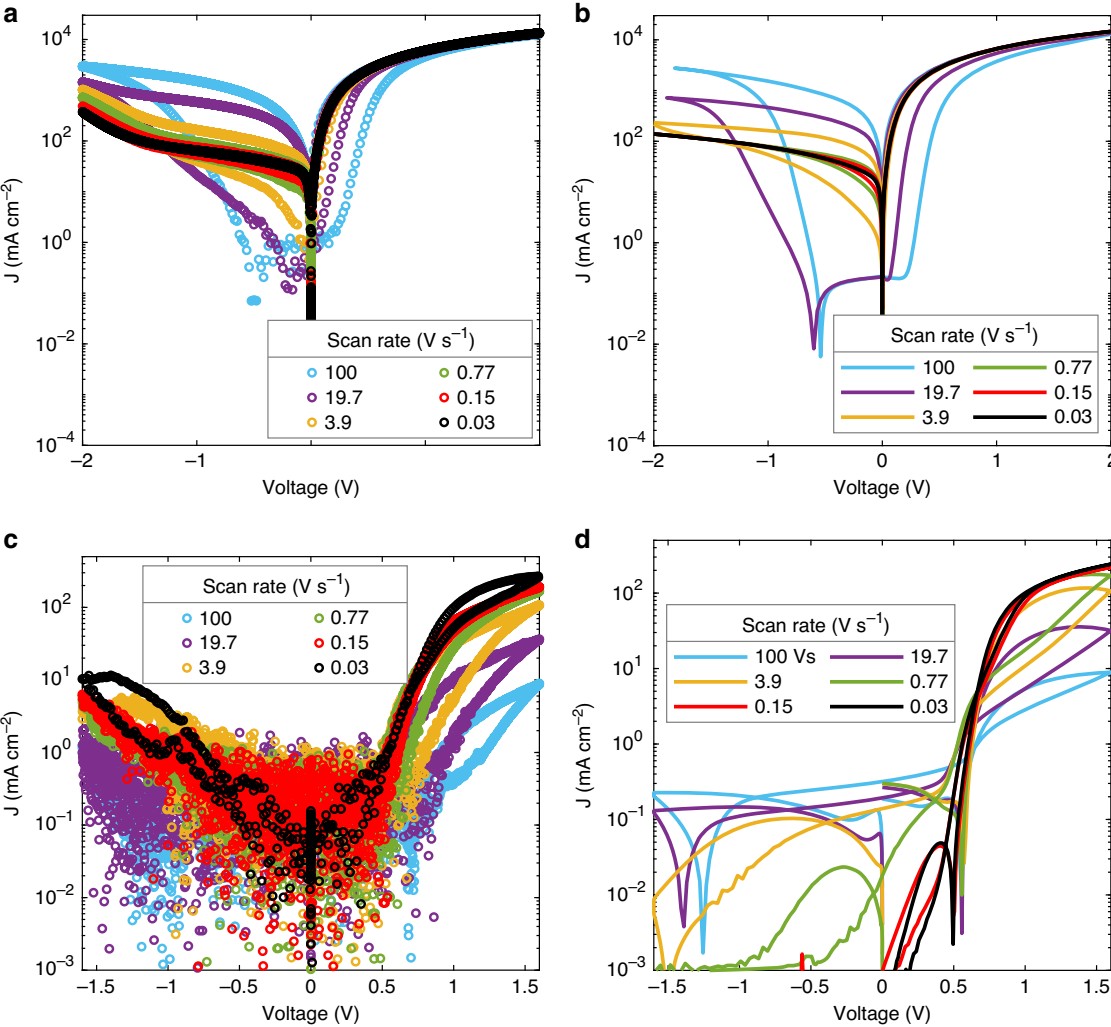

**Fig. 3 Current density–voltage characteristics of single-carrier devices for different voltage scan rates.** The measurements (symbols) for the electron-only (**a**) and hole-only (**c**) device are reproduced by the corresponding simulations (solid lines) in panel (b) and (d), respectively. The hole-only device in an Au/MAPbI$_3$/PTAA/Au configuration shows slightly asymmetric characteristics. The simulations incorporate a scan rate-dependent dielectric constant, resulting in a correct description of both the magnitude of the current and the hysteresis behavior.

in Supplementary Fig. 6, by assuming a lower value of the permittivity the shape of the J–V characteristics cannot be reproduced, irrespective of the chosen mobility.

**Scan-rate dependence of the SCLC.** To further confirm whether the frequency-dependent permittivity should be used in the SCLC analysis, we have performed current–voltage measurements as a function of scan rate, as displayed in Fig. 3a, c. As shown in Fig. 3b, d the scan-rate dependence of the current–voltage characteristics can only be completely reproduced by considering the frequency dependence of the permittivity for every scan rate (red stars in Fig. 1a). Although the electron current in forward bias is limited by the electrode series resistance at higher applied voltages, it is observed that the hole current in the hole-only device increases with decreasing scan rate, in accordance with the higher permittivity measured at lower frequencies. This increase cannot be reproduced by assuming a constant, scan-rate independent permittivity.

**Temperature dependence of the SCLC.** The relative permittivity also controls the temperature dependence of the electron and hole currents. As displayed in Fig. 1b, the quasi-static dielectric constant at a frequency of 0.0575 Hz decreases with decreasing temperature, which is associated with a decrease in ion diffusivity, as shown in Fig. 1c. Both these quantities are determined by impedance spectroscopy on MAPbI3 capacitors. The temperature-dependent dielectric constant directly results in a temperature dependence of the SCLCs, as displayed in Fig. 4. The device simulations correctly reproduce the temperature dependence, using the measured temperature-dependent permittivity (Fig. 1b) as input, while keeping the charge-carrier mobility constant. In addition, the hysteresis behavior is also correctly reproduced by the model, using the temperature-dependent diffusivity as shown in Fig. 1c. With regard to the hole-only current at low temperatures the ions are so slow that they cannot follow the voltage scan, resulting in the absence of hysteresis. With increasing temperature the ions become more mobile and the hysteresis increases. At room temperature, the further increased ion mobility reduces the hysteresis again, since the ions are sufficiently fast to follow the applied voltage signal. The hysteresis behavior of the electron-only is more complex due to the presence of a built-in voltage. The built-in voltage leads to diffusion of ions, in this case positive ions to the exiting contact, already at

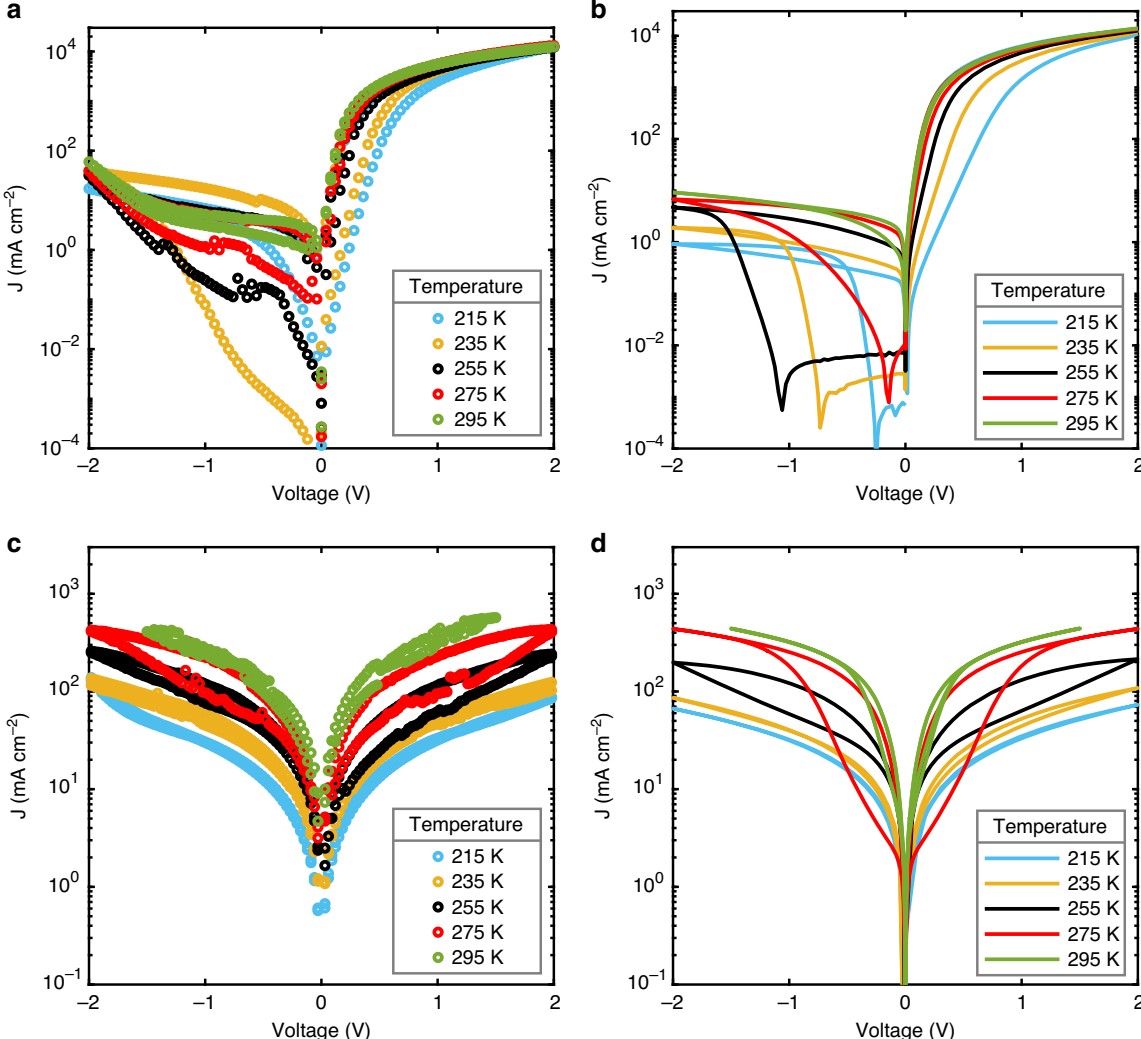

**Fig. 4 Temperature-dependent current density–voltage characteristics of single-carrier devices.** The electron (**a**) and hole (**c**) currents (symbols) were measured from 295 K to 215 K at a scan rate of 0.46 V s⁻¹. The simulations (solid lines) of the electron-only (**b**) and hole-only (**d**) device reproduce the experiments by using a temperature-dependent dielectric constant and ion diffusion coefficient, using the values as displayed in Fig. 1b, c.

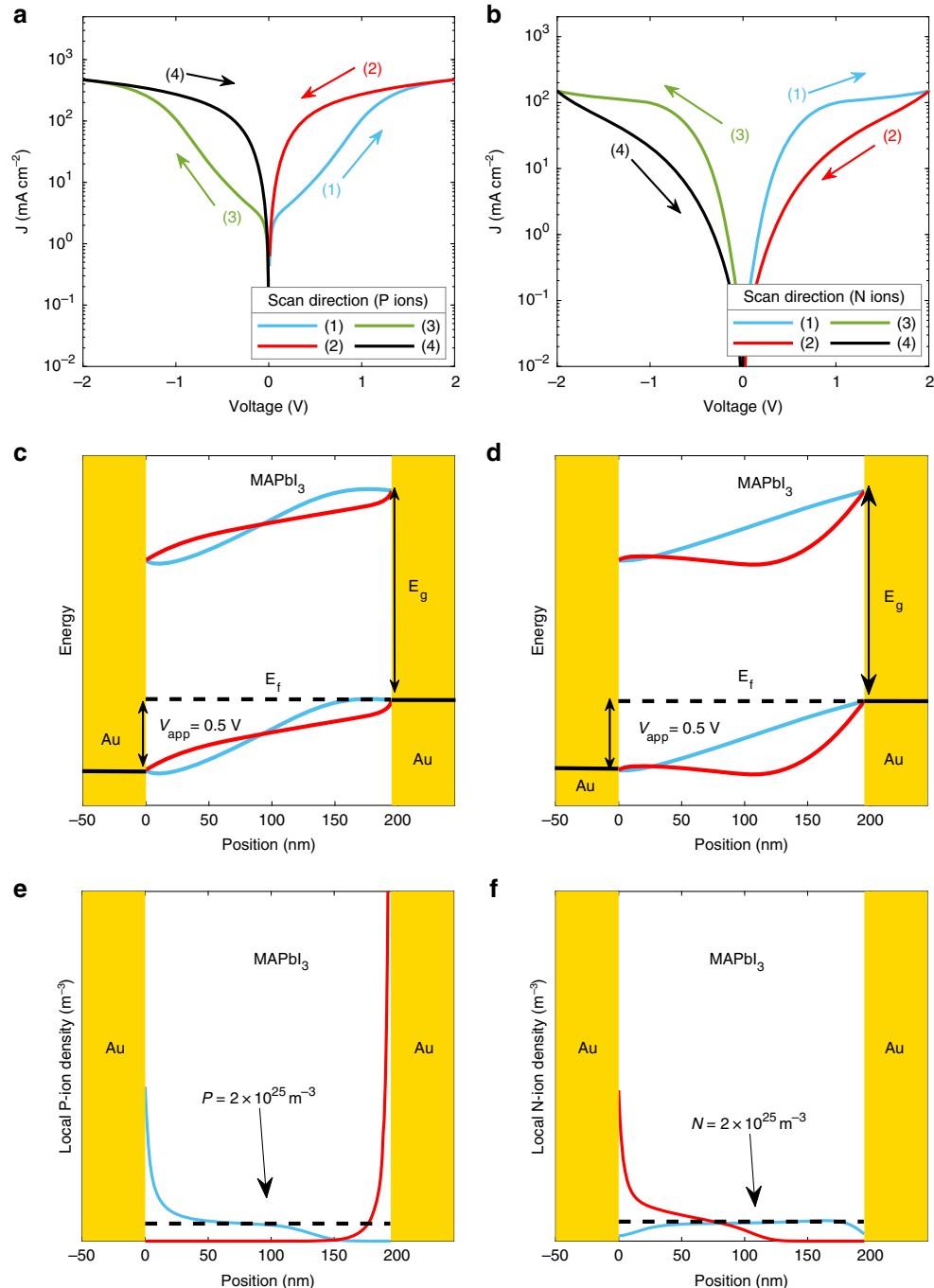

**Fig. 5 Current density–voltage characteristics of a hole-only device at 275 K.** The current density–voltage characteristics (solid lines) are simulated with the same set of parameters under two different conditions for ionic charges: mobile positive ions and a uniform density of immobile negative ions (**a**) and mobile negative ions and immobile positive ions (**b**). The corresponding energy band diagrams and ion distributions for mobile positive ions (**c**, **e**) as well as mobile negative ions (**d**, **f**) are displayed at a forward bias of 0.5 V for the up scan (blue solid lines) and down scan (red solid lines). The dashed lines in **e** and **f** indicate the average ion densities.

zero applied bias. At low temperatures, the immobile ions stay at this contact, independent of the applied voltage, which can give rise to negative electric fields in the device and large hysteresis. The effect of ion movement on the hysteresis will be discussed in more detail below. We note that a sharp increase is present in the experimental injection-limited current in high reverse bias, especially noticeable in the electron-only devices at higher temperatures. This might be associated with field-assisted charge injection, which will be enhanced by the accumulation of ions at the contact.

The accurate agreement between our experiments and simulations, based on experimentally determined parameters, demonstrates that the SCLC behavior in perovskites is completely governed by the temperature dependence of ion dynamics and the associated temperature and frequency-dependent permittivity. We note that temperature-dependent mobilities have been reported using different techniques, but in all cases the frequency and temperature dependence of permittivity due to ion dynamics were not taken into account in the interpretation of the data[24,28,29,53,64,65,69–72]. We cannot fully exclude a very small

temperature dependence of the charge-carrier mobility but, if present, is fully overwhelmed by the temperature dependence of the permittivity and therefore of no relevance for the description of the SCLC.

**Influence of ion motion on SCLCs.** Thus far, we have assumed that only positive ions are mobile in the simulations, which excellently reproduces the direction of hysteresis in the experimental data, as displayed in Fig. 5a. Here, the arrows and numbers represent the voltage scan direction. A question is whether this assumption also has an effect on the analysis of the SCLCs. For this reason, we have also performed simulations by assuming negative ions to be mobile, but keeping all other parameters the same. As a first case, we consider the hole transport at 275 K, where the hysteresis is maximal. Strikingly, as shown in Fig. 5b, the direction of the hysteresis is reversed when assuming mobile negative ions. A similar observation is obtained for electron-only devices (Supplementary Fig. 8). Also here, only mobile positive carriers provide the correct direction of the hysteresis. We note that for the electron-only devices, we modeled experiments at higher scan rate, as the hysteresis is more pronounced in that case.

To explain this behavior in more detail, the energy band diagrams of the hole-only device at an applied voltage of 0.5 V are displayed in Fig. 5c, d. Consider the example of a hole-only device with positive ions. On the up scan (from 0 to 2 V, forward bias), the positive ions will migrate in the direction of the negatively biased (right) electrode, at which the holes are extracted. As these ions move slowly, the accumulation of ions at the extracting electrode at a bias voltage of 0.5 V is still rather limited (Figs. 5 (1)), having a minor influence on the electric-field distribution across the device (Fig. 5c). However, on the down scan (Figs. 5a (2)), the positive ions have migrated further away from the positively biased electrode (Fig. 5c), resulting in positive ion depletion and an associated positive field near the hole injection electrode, enhancing the injection of holes. This results in a higher current in the down scan (Figs. 5a (2). In the case of negative ions, the ions migrate toward the positively biased, hole injecting electrode, where they screen the applied field (Fig. 5b (1)). Also in this case, the ions are distributed relatively uniformly on the up scan (Fig. 5d), whereas the field screening limits hole injection on the down scan and, in turn, a decreased hole current is observed (Fig. 5b (2)). The corresponding electric field, hole, and ion concentration profiles for both cases are shown in Supplementary Figs. 9 and 10, respectively. As a result, the direction of the hysteresis in MAPbI$_3$ single-carrier devices directly reveals the sign of the moving ionic species.

## Discussion

In conclusion, we have demonstrated the importance of temperature-dependent ion dynamics and a temperature- and frequency-dependent apparent dielectric constant on the analysis of SCLCs in hybrid organic–inorganic perovskites. We have developed a device model that can reproduce the scan-rate and temperature-dependent current–voltage characteristics with only experimentally validated parameters as input. The sign of the dominant mobile ionic species can be determined from the direction of the hysteresis in electron- and hole-only devices. The basic understanding of electron and hole currents in perovskite layers is an important step in the direction of unraveling the device physics of perovskite solar cells and light-emitting diodes, in which charge recombination is an additional factor that needs to be taken into account. Only by successive experimental validation of separate factors, such as ion dynamics and charge

transport, perovskite device models can be built up with increasing complexity, containing a large body of input variables.

## Methods

**Device fabrication.** For fabricating electron-only (hole-only) devices, layers of Cr/Ag (Cr/Au) (1 nm/60 nm) was deposited as the bottom contact on a glass substrate by thermal evaporation in high vacuum. For Au/PTAA/MAPbI$_3$/Au hole-only devices, a thin layer (10 nm) of PTAA was spin coated from a solution of PTAA in toluene (2 mg/mL) on the bottom Au electrode. A 30 wt.% solution of Methylammonium iodide and lead acetate trigydrate (PbAc$_2$) with a molar ratio of 3:1 in N,N-dimethylformamide was prepared and spin coated on the substrate inside a nitrogen-filled glovebox[12]. The MAPbI$_3$ films were annealed at 100 °C for 30 min on a hot plate. The hole-only devices were finished by thermally evaporating a layer of Au (60 nm) as top electrode. For electron-only devices, 5 nm of C$_{60}$ and 5 nm of TPBi were thermally evaporated to achieve efficient electron injection into the MAPbI$_3$ layer. On top of that, a capping layer of Al (100 nm) was deposited by thermal evaporation. The area of the devices was 1 mm$^2$.

**Characterization.** Impedance measurements were performed using a computer-controlled Solartron impedance analyzer. The current–voltage measurements were carried out using a Keithley 2400 source meter. The scan-rate-dependent measurements were conducted using a Paios system from Fluxim. All the measurements were performed inside a nitrogen-filled glovebox.

**Simulation.** Device simulations were performed using an experimentally validated electronic-ionic drift-diffusion model. For solving the drift-diffusion equations coupled with Poisson's equation, a code was programmed in MATLAB. The model includes both electronic and ionic conduction. The program numerically solves the one-dimensional drift-diffusion equations for electrons, holes, and ions by forward integration over time and space. The ions are confined inside the perovskite layer by the ion-blocking contacts. The input ionic properties were experimentally quantified by impedance spectroscopy and electric displacement measurements[18]. For modeling the single-carrier devices, the injection barrier for the majority carriers is taken to be zero and the injection of the minority carriers was minimized by applying large injection barriers. Current–voltage calculations were conducted with the same protocol as the experimental measurements, including the voltage scan range and scan rate, as well as the temperature. The effects of the electrode series resistance on the electron-only data were included in the simulation results.

## Data availability

Experimental data and simulation code are available from the corresponding author upon reasonable request.

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

## Acknowledgements

We thank Christian Bauer, Michelle Beuchel, Hans-Jürgen Guttmann, Frank Keller, and Verona Maus for technical support. Open access funding provided by Projekt DEAL.

## Author contributions

G.A.H.W. and P.W.M.B. proposed the project. M.S.A. fabricated the devices and performed all measurements. M.S.A. programmed the drift-diffusion model and performed the simulations. All authors contributed to the experimental and theoretical analysis. P.W.M.B. and G.A.H.W. supervised the project and all authors contributed to preparing and writing of the manuscript.

## Competing interests

The authors declare no competing interests.
