## [Peer Review File · Nature Communications]

REVIEWER COMMENTS

Reviewer #1 (Remarks to the Author):

This paper analyses current-voltage curves in terms of space charge limited currents. This method is popular for deducing charge transport parameters in organic semiconductors but I am not aware of its being used before for perovskite solar cells. With the aid of a drift diffusion model that allows for mobile ions that greatly modify perovskite cell current-voltage characteristics, and by fabricating electron and hole only devices, the authors have been able to find electron and hole mobilities I believe for the first time. Many useful experimental results are presented in addition to the mobilities such as the temperature dependence of the permittivity.

I recommend the paper is published ****provided**** more explanation is provided on the materials science underlying the results obtained from experiment and model. I refer to

- a) the reason why the electron mobility exceeds the hole mobility and how sample dependent this result is.
- b) the origins of the observed frequency dependence of the permittivity
- c) identification of the dominant mobile ionic species

Reviewer #2 (Remarks to the Author):

Review for the manuscript "Space-charge-limited electron and hole currents in hybrid organic-inorganic perovskites" by Alvar et al.

The paper reports on the measurement and simulation of unipolar halide perovskite devices with the purpose of accurately determine the electron and hole mobility by correctly accounting for ionic motion. This is done by implementing ionic motion in a drift-diffusion solver and then fitting or reproducing the experimental data with that simulation tool.

The general idea is very nice and timely. But I do have some technical questions and issues that I don't understand that could be quite important for the complete story.

The authors highlight the necessity to include the frequency dependent permittivity, especially the low frequency part influenced by ions. However, if I understand correctly the extremely high permittivity is a result of the ion movement and therefore already part of the simulation. That means if we run the simulation with the permittivity shown in Fig. 1a, we would be counting the effect of ions twice. This confuses me a lot. Could you clarify? If you were to simulate the plate capacitance (not sure this is possible with the simulation tool) of a sample with ion movement, would it show the trend shown in Fig. 1a for a (nearly) constant permittivity used in the simulation? This is what I understood should happen.

The second big issue that I have are the low mobilities. First, the unit of mobility is cm^2/Vs and the authors have forgotten the V. I briefly thought they may be reporting diffusion constants rather than mobilities but I assume it is just a missing V in the unit. Assuming this is correct, the mobilities are very low, the hole mobility being lower than that of typical organic semiconductors. If that is correct, something is really odd either with the samples or with the analysis method. Let us assume the hole mobility is on the order of $1\text{e-}6 \text{ cm}^2/\text{Vs}$, then the diffusion length even for a high bulk lifetime of $1\mu\text{s}$ would be on the order of 3nm. Solar cells are typically 300-800nm thick and usually show little signs of charge collection problems and also little signs of imbalanced collection. How can this be compatible? Did the authors ever try to make a solar cell from their MAPI and see how efficiency and charge collection are like and then simulate it and see whether it is all self-consistent? at the moment, it seems not self-consistent.

There is also a statement saying "We note that our extracted mobilities are comparable to values obtained by electrode-based mobility measurement techniques^{31, 40, 43, 63, 64} and lower than the ones obtained by electrode-free techniques^{27, 28, 31}."

I didn't check all the references, but only the first one (31). Here, I note that the range of values reported is huge going from 10^{-5} to 10^3 cm²/Vs. It is not credible to me that the whole range is correct and it is only a matter of what subset of the carrier population is probed. I also don't believe that this is sample to sample variation. I think the only credible explanation for this is that in some cases, the mobility measurement doesn't measure mobility but something else or that the analysis method that translates experimental results into mobilities is flawed. I also note that SCLC (an electrode based technique) is listed rather in the higher range of values. Same holds for Hall, also a technique with electrodes. This contradicts the author's statement that suggests that such low mobilities are the "normal" scenario for electrode based techniques. I think they are the "normal" scenario for wrong data analysis. The latter sounds harsh but is just based on me believing in the fact that mobilities have to be consistent with e.g. photovoltaic behaviour. If a solar cell works, its mobility and lifetime cannot be infinitely low.

The mobility we would need to learn something for perovskite based devices is a mobility that helps explain functionality within a sensible model. These low mobilities in my opinion do not help to explain functionality but I am open to hear the opinion of the authors. But currently, I have the feeling that whatever this number really means it won't help without a lot of further explanation.

Reviewer #3 (Remarks to the Author):

The authors present an interesting results on demonstration of space charge limited current with respect to temperature dependence, scan rate dependence, and frequency dependence. The correlation between experimental data and simulation well supports the hypothesis and conclusion of this work. Also the manuscript is well written. All the current voltage measurements were performed as staircase measurement and measurement history is not provided. As concerned in the research society, in the literature and even by the author, the hysteresis originated by ion movement is a main issue to understand the space charge limited current. I do believe this work will give a direction of unraveling the device physics of perovskite based optoelectronic devices, after the authors provide more details about reproducibility/measurement history and also address my following concerns.

1. Line 75: how do you know this high dielectric constant is a result of slow moving ions? What is the timescale at which ions move?
2. Line 87: Is it still an ohmic contact with having organic semiconductor layer even though it is thin layers? Can author confirm that this asymmetric current voltage is only attributed to the electron injection barrier at the Ag bottom electrode? Can author explain or confirm that is not due to the C60 layers on top of perovskite?
3. Line 111: on the same note to above question 2, Can author explain what is the reason for this assumption? It was also not clearly described in ref 17 (doi.org/10.1002/aelm.201900935).
4. Line 125: Which factor causes the ' $\epsilon \times V_m$ '? In this frequency rate indeed the ions are moving, which causes a change in the dielectric constant, but that also means the JV curves of the sclc measurement are strongly affected, which could make it arguable to be able to fit these JV curves.
5. Line 132: With organic semiconductor layer on the bottom metal electrode (PTAA) in comparison to direct spin coat on metal electrode, the morphology and crystallinity, grain size of perovskite thin films

should be confirm if they are still the same perovskite, like via XRD or SEM. What is the thickness of perovskite thin-films?

6. Fig. S2: JV curves are staircase measurement, which means the measurement is conducted under a constantly applied electric field. It doesn't provide an information about the history of the measurement. Therefore the ions might have already moved or are moving during the scan, and thereby affecting the extracted current (both the shape of JV curves and the magnitude).

7. Line 135-140: The SCLC part has slope equal 2. However, since 1) the ions are affecting the JV curve (and this influencing the slope) and 2) The SCLC regime has not yet been reached, its hard to validate if the author can actually extract an effective mobility from there. For that to be possible to slope has to be fitted, for which the effect of ion density (and their movement in this case) has to be known.

8. Fig S3: It might be a good plan to perform a pulsed measurement and perform the fitting with low dielectric constant. If that would fit it would be the same as the fast measurement with high dielectric constant. However, it seems impossible to set 1 single value (the dielectric constant) that has to take into account all the possible effects from ions. (proven by the observed hysteresis).

9. Line 177: What is the mechanism behind an increase ion mobility causing enhancement of the screening of the electric field, but thereby reducing the hysteresis.

10. Line 207: Fig 4 needs to be Fig. 5

11. Line 213: Figs 5c and 5d, need to be 5a and 5b.

12. In Fig. 2 for electron only device, there is a rapid increase in current near at -2V but not in hole only devices. However it's found in the Fig 3a for electron only device that the rapid increase is getting effective as scan rate decrease. Moreover these rapid changes are shown in both electron only and hole only device but in opposite temperature. It is very hard to understand this characteristic curve with respect to neither temperature dependent, electron only/hole only, nor scan rate. I believe that the author should note this point in the main text.

REVIEWER COMMENTS (please find our response to all reviewer comments in blue)

Reviewer #1 (Remarks to the Author):

This paper analyses current-voltage curves in terms of space charge limited currents. This method is popular for deducing charge transport parameters in organic semiconductors but I am not aware of its being used before for perovskite solar cells. With the aid of a drift diffusion model that allows for mobile ions that greatly modify perovskite cell current-voltage characteristics, and by fabricating electron and hole only devices, the authors have been able to find electron and hole mobilities I believe for the first time. Many useful experimental results are presented in addition to the mobilities such as the temperature dependence of the permittivity.

We thank the reviewer for these positive comments.

I recommend the paper is published ****provided**** more explanation is provided on the materials science underlying the results obtained from experiment and model. I refer to:

a) The reason why the electron mobility exceeds the hole mobility and how sample dependent this result is.

At this point, from these results we cannot clarify the reason for unbalanced charge transport. However, our results show that this behavior is highly reproducible over numerous samples devices that we have fabricated and tested over the last three years, as now indicated in the manuscript. As we pointed out in the manuscript, shallow trapping on defects could be the origin of the lower hole transport. However, this cannot be verified with SCLC measurements. We also do not exclude that the charge balance may be different for different perovskite compositions, or, possibly, preparation methods. The main purpose of this manuscript is to show how ion motion modifies the interpretation of SCLC in mixed electronic-ionic semiconductors.

b) The origins of the observed frequency dependence of the permittivity

The origin of the frequency dependence of the permittivity has been described in our previous study, Ref. 18, in which by impedance analysis we showed that the slow mobile ionic species lead to an increase in the permittivity at low frequencies. We have now referred to this study more clearly. We note that Ref. 18 may not have been published by the time of first submission, for which we apologize. In Ref. 18 we have also presented a model [Ref. 18, Eq. (8)] showing how the permittivity depends on frequency as a result of ion motion.

c) Identification of the dominant mobile ionic species

In our previous report (Ref. 18), where we presented our mixed electronic-ionic device model, we made an assumption that the mobile ionic charges are positively charged. In the present manuscript, as discussed along Figure 5, we confirm that the mobile ionic species are indeed positively charged, based on the direction of the hysteresis. Although candidates such as iodine vacancies and methylammonium ions have been put forward as mobile positively charged ions in literature, from our electrical characterization the nature of mobile ionic species cannot be recognized and is out of the scope of this study. However, the diffusion coefficient would correspond to either methylammonium ions or iodine vacancies, considering reported diffusion coefficients in literature [Refs. 15, 37, 63]. We have added this to the manuscript.

Reviewer #2 (Remarks to the Author):

Review for the manuscript "Space-charge-limited electron and hole currents in hybrid organic-inorganic perovskites" by Alvar et al. The paper reports on the measurement and simulation of unipolar halide perovskite devices with the purpose of accurately determine the electron and hole mobility by correctly accounting for ionic motion. This is done by implementing ionic motion in a drift-diffusion solver and then fitting or reproducing the experimental data with that simulation tool. The general idea is very nice and timely. But I do have some technical questions and issues that I don't understand that could be quite important for the complete story.

We thank the reviewer for these positive comments.

The authors highlight the necessity to include the frequency dependent permittivity, especially the low frequency part influenced by ions. However, if I understand correctly the extremely high permittivity is a result of the ion movement and therefore already part of the simulation. That means if we run the simulation with the permittivity shown in Fig. 1a, we would be counting the effect of ions twice. This confuses me a lot. Could you clarify?

In Ref. 18, by electric displacement-voltage measurements at different frequencies and simulating the experimental results with our electronic-ionic device model, we have shown that the frequency dependence of electric displacement can only be explained by a frequency-dependent permittivity. Figure S4 in Ref. 18 shows that simulating the frequency dependent electric displacement cannot be modeled with constant value of permittivity for all the frequencies. While the ion movement is incorporated in the drift-diffusion simulations, it does not automatically result in the high measured capacitance, for which the permittivity has to be increased. We have added this to the manuscript.

Additionally, in the same report [Ref. 18] the effect of ion diffusion has been studied and distinguished from the effect of permittivity. In the simulations, the permittivity is responsible for the distribution of charges in the device and the ion diffusion controls the variation of this charge distribution.

If you were to simulate the plate capacitance (not sure this is possible with the simulation tool) of a sample with ion movement, would it show the trend shown in Fig. 1a for a (nearly) constant permittivity used in the simulation? This is what I understood should happen.

As stated above, we have simulated the electric displacement behavior of a capacitor (=the amount of free charges on the plates at a given applied voltage) in Figure 4 of Ref. 18 by taking into account the frequency dependent permittivity. By definition, the amount of charges on the plate divided by the applied voltage represents the plate capacitance. As shown in Figure S4 taking a constant permittivity strongly underestimates the electric displacement response and resulting capacitance at low frequencies.

The second big issue that I have are the low mobilities. First, the unit of mobility is cm^2/Vs and the authors have forgotten the V. I briefly thought they may be reporting diffusion constants rather than mobilities but I assume it is just a missing V in the unit.

We apologize for this mistake. Indeed the reported quantity is the mobility ($\frac{m^2}{V.s}$), not the diffusion coefficient. We thank the reviewer for noticing this error and we have corrected it accordingly.

Assuming this is correct, the mobilities are very low, the hole mobility being lower than that of typical organic semiconductors. If that is correct, something is really odd either with the samples or with the analysis method. Let us assume the hole mobility is on the order of $1\text{e-}6\text{ cm}^2/\text{Vs}$, then the diffusion length even for a high bulk lifetime of $1\mu\text{s}$ would be on the order of 3nm . Solar cells are typically $300\text{-}800\text{nm}$ thick and usually show little signs of charge collection problems and also little signs of imbalanced collection. How can this be compatible? Did the authors ever try to make a solar cell from their MAPI and see how efficiency and charge collection are like and then simulate it and see whether it is all self-consistent? at the moment, it seems not self-consistent. There is also a statement saying "We note that our extracted mobilities are comparable to values obtained by electrode-based mobility measurement techniques^{31, 40, 43, 63, 64} and lower than the ones obtained by electrode-free techniques^{27, 28, 31}." I didn't check all the references, but only the first one (31). Here, I note that the range of values reported is huge going from 10^{-5} to $10^3\text{ cm}^2/\text{Vs}$. It is not credible to me that the whole range is correct and it is only a matter of what subset of the carrier population is probed. I also don't believe that this is sample to sample variation. I think the only credible explanation for this is that in some cases, the mobility measurement doesn't measure mobility but something else or that the analysis method that translates experimental results into mobilities is flawed. I also note that SCLC (an electrode based technique) is listed rather in the higher range of values. Same holds for Hall, also a technique with electrodes. This contradicts the author's statement that suggests that such low mobilities are the "normal" scenario for electrode based techniques. I think they are the "normal" scenario for wrong data analysis. The latter sounds harsh but is just based on me believing in the fact that mobilities have to be consistent with e.g. photovoltaic behaviour. If a solar cell works, its mobility and lifetime cannot be infinitely low. The mobility we would need to learn something for perovskite based devices is a mobility that helps explain functionality within a sensible model. These low mobilities in my opinion do not help to explain functionality but I am open to hear the opinion of the authors. But currently, I have the feeling that whatever this number really means it won't help without a lot of further explanation.

The reviewer raises a good point regarding the low hole mobility and the apparent discrepancy with the generally good performance of MAPbI_3 solar cells. Let us first start with an expression for the drift-diffusion photogenerated current in an insulator as derived by Sokel and Hughes [J. Appl. Phys. 53, 7414 (1982)], in the absence of recombination:

$$J = eGL \left(\frac{\exp\left(\frac{eV}{kT}\right) + 1}{\exp\left(\frac{eV}{kT}\right) - 1} - \frac{2kT}{eV} \right)$$

This equation shows that the mobility is not important in case recombination is absent in a solar cell. Of course we do not intend to state that recombination is absent in a perovskite solar cell, but it illustrates that a high mobility is not required when recombination rates are low.

We have fabricated solar cells with our MAPbI_3 thin films in the inverted planar structure of ITO/PTAA(8 nm)/ MAPbI_3 (250 nm)/ C_{60} (5 nm)/TPBi(5 nm)/Al(100 nm). The J - V characteristic of the device is presented in Supplementary Figure S7 and below. The J - V of the device is simulated using our electronic drift-diffusion model with the *measured* electron and hole motilities. Here the recombination is assumed to be of the Langevin type with a Langevin reduction prefactor of 0.01. A similar Langevin prefactor has been used previously in device simulations of perovskite solar cells (Adv. Energy Mater.

2017, 7, 1602432), when comparing the used bimolecular recombination rate to the calculated Langevin rate. This proves that even with such low hole mobility the device characteristics can be simulated because of low recombination rates. We note that even lower Langevin prefactors have been determined in experimental studies (C. Wehrenfennig, *Adv. Mater.*, 2014, 26, 1584–1589). More extensive solar-cell simulations will be the subject of future study.

Concerning the mobility values from different methods, we believe that the large discrepancy in the reported values comes from both the measurement method, its interpretation, and the sample (e.g. morphology). It is expected that single crystal samples give higher mobility values due to their defect-free crystal structure in comparison to thin films. Furthermore, the method to obtain the mobility can play an important role. For instance, performing Terahertz measurements on a single grain in a thin film would result in high values of mobility, due to exclusion of the effect of the grain boundaries on charge transport.

It has to be noted that the SCLC and the Hall mobilities that are mentioned in the references are measured on MAPbI₃ single crystals. Compared to single crystals, thin films have several imperfections in their structure and morphology which likely limit the charge transport.

More importantly, it has to be noted that in the references that report SCLC mobilities, the classical SCLC analysis has been used, which is not applicable to such an electronic-ionic systems and results in overestimation of the mobility.

In this manuscript our objective is to provide a better understanding of the interpretation of SCLC measurements on perovskite devices and to point out the fact that application of classical SCLC analysis to perovskite devices results in overestimation of the mobility.

Reviewer #3 (Remarks to the Author):

The authors present an interesting results on demonstration of space charge limited current with respect to temperature dependence, scan rate dependence, and frequency dependence. The correlation between experimental data and simulation well supports the hypothesis and conclusion of this work. Also the manuscript is well written. All the current voltage measurements were performed as staircase measurement and measurement history is not provided. As concerned in the research society, in the literature and even by the author, the hysteresis originated by ion movement is a main issue to understand the space charge limited current. I do believe this work will give a direction of unraveling the device physics of perovskite based optoelectronic devices, after the authors provide more details about reproducibility/measurement history and also address my following concerns.

We thank the reviewer for these positive comments.

1.Line 75: how do you know this high dielectric constant is a result of slow moving ions? What is the timescale at which ions move?

In Ref. 18 using impedance analysis we showed that the frequency dependence of the permittivity is a consequence of ion motion. We have also presented an equation [Ref. 18, Eq (8)] for the frequency dependence of the permittivity in presence of mobile ions. We have now referred to Ref. 18 at this point.

From the same study, the diffusion coefficient of ions in our MAPbI₃ thin films was deduced to be $10^{-15} \text{ m}^2/\text{s}$. This corresponds to ion movement on the ~second time scale.

2. Line 87: Is it still an ohmic contact with having organic semiconductor layer even though it is thin layers? Can author confirm that this asymmetric current voltage is only attributed to the electron injection barrier at the Ag bottom electrode? Can author explain or confirm that is not due to the C60 layers on top of perovskite?

Reference 62 shows that the electron affinity of C₆₀ and MAPbI₃ align very well, so a barrier is not expected at this interface. The electron-injection barrier at the Ag bottom electrode is expected to be much larger in comparison, considering the Ag work function of 4.6 eV. Therefore, this is the likely origin for the asymmetry in the current-voltage characteristics.

To verify this experimentally, we have fabricated electron-only by replacing the Ag bottom electrode by Au, which has a higher work function. Therefore, the electron-injection barrier and the concomitant asymmetry in the J-V characteristics should be larger. The current-voltage characteristics of the Au/MAPbI₃/C₆₀/TPBi/Al and Ag/MAPbI₃/C₆₀/TPBi/Al devices are compared in Supplementary Figure S4. The asymmetry clearly increases by using the higher work function Au electrode, demonstrating that the work function of the metal bottom electrode controls the asymmetry.

3. Line 111: on the same note to above question 2, Can author explain what is the reason for this assumption? It was also not clearly described in ref 17 (doi.org/10.1002/aelm.201900935).

The assumption of mobile positive ionic charges and immobile negative ions in our recent study [now Ref. 18] initially arose from literature. Here, as discussed along Figure 5, this assumption is confirmed by investigating the hysteresis in single carrier current-voltage we showed that the assumption of positive mobile ionic charges is correct. We have now explained that we based this on previous studies and identified candidates for the positive ionic species.

4. Line 125: Which factor causes the '4 x V_m'? In this frequency rate indeed the ions are moving, which causes a change in the dielectric constant, but that also means the JV curves of the sclc measurement are strongly affected, which could make it arguable to be able to fit these JV curves.

In our current voltage measurements, we use a ramp voltage with an amplitude V_m and a frequency f . Considering the period of the applied signal $T = 1/f$, the scan rate of the ramp would be $scan\ rate = \frac{V_m}{T/4}$. In other words, the period of the signal at different scan rates equals $T = \frac{4V_m}{scan\ rate}$ which corresponds to a frequency of $f = \frac{scan\ rate}{4V_m}$. In this way the scan rate of current-voltage measurements is translated into the frequency. We have added an explanatory figure to the SI.

5. Line 132: With organic semiconductor layer on the bottom metal electrode (PTAA) in comparison to direct spin coat on metal electrode, the morphology and crystallinity, grain size of perovskite thin films should be confirm if they are still the same perovskite, like via XRD or SEM. What is the thickness of perovskite thin-films?

The SEM images are added to the SI. The grain size and shape look rather similar.

The thickness of the films in these devices is 195 nm.

6. Fig. S2: JV curves are staircase measurement, which means the measurement is conducted under a constantly applied electric field. It doesn't provide an information about the history of the measurement. Therefore the ions might have already moved or are moving during the scan, and thereby affecting the extracted current (both the shape of JV curves and the magnitude).

For this reason we have performed the scan rate dependent measurement, which allows to study the behavior of the device at different scan rates.

It should be noted that the same measurement protocol has been implemented in the simulation, i.e. ions are moving during the scan (which is rate dependent) both in simulation and experiment.

7. Line 135-140: The SCLC part has slope equal 2. However, since 1) the ions are affecting the JV curve (and this influencing the slope) and 2) The SCLC regime has not yet been reached, its hard to validate if the author can actually extract an effective mobility from there. For that to be possible to slope has to be fitted, for which the effect of ion density (and their movement in this case) has to be known.

Here we discussed the effect of shallow trapping on the space-charge-limited current. To illustrate this, we referred to the classical SCLC equations as derived for intrinsic semiconductors without ions, to show that shallow trapping modifies the effective mobility, while not changing the other features of SCLC. However, the reviewer is correct that ions do affect the slope of the J-V. However, this does not mean that the current is only space-charge limited when it has a slope of 2. The ions modify the charge distribution, which results in an SCLC with a slope of less than 2. The simulations reproduce this behavior, which means that we can extract a mobility – or effective mobility – even when the slope is not 2. To avoid confusion, we have rephrased this section.

8. Fig S3: It might be a good plan to perform a pulsed measurement and perform the fitting with low dielectric constant. If that would fit it would be the same as the fast measurement with high dielectric constant. However, it seems impossible to set 1 single value (the dielectric constant) that has to take into account all the possible effects from ions. (proven by the observed hysteresis).

In principle, we have done this with a fast scan rate measurement, in which the dielectric constant is low. However, here we have the problem that the current is limited by the electrode series resistance (which we have tried to reduce as much as possible in our device layout) and thus we cannot measure higher currents. As correctly noted by the reviewer, considering the scan rate dependent hysteresis, it is impossible to explain the I-Vs at all scan rates with a single value for the permittivity.

9. Line 177: What is the mechanism behind an increase ion mobility causing enhancement of the screening of the electric field, but thereby reducing the hysteresis.

Due to the increased ion mobility the ions are fast enough to follow the applied voltage signal completely, resulting in the absence of hysteresis. With enhanced screening we meant that faster ions result in higher permittivities and thus more screening, resulting in weaker current response to voltage changes. We have now rephrased this and focus only on the faster ions being able to follow the voltage signal, being the main effect.

10. Line 207: Fig 4 needs to be Fig. 5

11. Line 213: Figs 5c and 5d, need to be 5a and 5b.

We thank the reviewer for noticing both these errors and we have corrected them accordingly. Please note that we do refer to the band diagrams in Figs. 5c and 5d here. However, we accidentally referred to the arrows and numbers in Fig. 5a and 5b directly afterwards. We have therefore moved this sentence up, when referring to Fig. 5a. We apologize for the confusion.

12. In Fig. 2 for electron only device, there is a rapid increase in current near at -2V but not in hole only devices. However it's found in the Fig 3a for electron only device that the rapid increase is getting effective as scan rate decrease. Moreover these rapid changes are shown in both electron only and hole only device but in opposite temperature. It is very hard to understand this characteristic curve with respect to neither temperature dependent, electron only/hole only, nor scan rate. I believe that the author should note this point in the main text.

The reviewer has observed this feature correctly. This rapid increase in the reverse bias current is not reproduced by the simulations. The reason for this is that the current in reverse bias is injection limited, since there exists a considerable barrier between the bottom electrode and the perovskite. To describe injection-limited currents correctly, an injection model has to be included. For instance, it might be that field-assisted charge injection, such as Fowler-Nordheim tunneling plays a role, especially considering the presence of ions. However, describing injection-limited currents correctly is even still a problem for organic semiconductors, which are far better understood than perovskites at the moment. Therefore, describing the injection-limited currents correctly is outside the scope of this paper. Therefore, we mainly focus on the space-charge-limited currents in forward bias. It should be noted that, apart from certain features at higher fields, the reverse bias currents are still fairly well described using standard Schottky diffusion theory, as can be seen in the figures. We have added a note to the paper regarding this current increase in reverse bias, as suggested by the reviewer.

REVIEWERS' COMMENTS:

Reviewer #1 (Remarks to the Author):

In this resubmission, the authors have answered the concerns raised by the reviewers. I am happy to recommend this version for publication in Nature Communications.

Reviewer #2 (Remarks to the Author):

Review for the revised version of the paper "Space-charge-limited electron and hole currents in hybrid organic-inorganic perovskites" by Alvar et al.

The key issue I had in the previous round was with the extremely low values of the mobility that do not seem to be consistent with any photovoltaic behaviour. In order to rationalize their findings, the authors have provided a simulated and experimental JV curve using Langevin-type recombination. This implies that they have set the SRH lifetime to infinity, i.e. assume that they are in the radiative limit. Unlike in organics, the bimolecular recombination coefficient found in literature is determined by radiative recombination and can therefore be calculated from the absorption coefficient and intrinsic carrier concentration of the material (see e.g. <https://www.nature.com/articles/s41467-017-02670-2>). Therefore, k_{rad} is known to a certain degree and doesn't depend on mobility. The papers cited by the authors (Sherkar et al. and Wehrenpfennig et al.) also find similar values.

If the typically cited range of mobilities of 1 - 100 cm²/Vs is correct, Langevin type recombination would be much faster than that and it makes no sense to assume that recombination was diffusion limited. Even if due to the presence of grain boundaries the external mobility was much slower (orders of magnitude than that), the Langevin theory should really use the local mobility that corresponds to what is determined by non-contact measurements. So I don't find it credible to use Langevin theory here.

The JV curve of the solar cell shows a Voc of ~1.05V for a MAPI cell. Thus, the cell has a Voc that is ~270mV away from the radiative limit (1.32 V for a band gap of 1.6eV). Thus, the non-radiative recombination current exceeds the radiative recombination current by a factor $\exp(270/25.8) = 3.5e4$. Under these circumstances it is most likely not appropriate to calculate the JV curve using what is essentially radiative recombination only. Certainly, part of the voltage loss will be due to interface recombination, part due to bulk recombination. How exactly the loss is distributed, we don't know. But my original argument still holds. The mobilities are too low to explain half way decent photovoltaic functionality in a realistic scenario (of finite SRH lifetimes that everybody reports who looks at transient optical measurements). The electron mobilities are on the edge of being just about credible if the lifetime of the bulk was very long. But the hole mobilities don't seem to be credible. What is causing them to be or appear so low, I don't know. Generally, to understand what the authors did in Fig. S7, I would need to see a full list of variables including all contact and interface layers. At the moment, I don't even understand how the curve can reproduce Voc while only having a Langevin recombination rate in the bulk. That should really give the radiative limit if correct values for band gap and effective DOS are used.

The general problem I have here is that to a certain degree I can only trust the authors that this is what their simulations are giving and there is no other way to simulate the data. As a reviewer (without access to the software and without the time to reproduce everything from scratch) I can only judge whether the data makes sense to me and it still doesn't. What will the publication of the article cause when it is read?

People who do SCLC on perovskites will see that they have to include ions in drift diffusion, which they probably already figured out themselves. Only problem is that 99% cannot do that because they don't have the code for it and no time to program one of their own. So those 99% only have the data as a

take home message and that will say what most of you thought the mobility of holes in MAPI is, is off by 6 to 8 orders of magnitude. What will the reader say? He/she will probably just be confused. As I am.

So, the take home message of the paper is not helpful to all those readers, who are not in the author's group and can use the code and I don't see that the paper develops in a direction where it would be helpful. Now one can of course still publish it because it contains interesting insights for the minority of people who can work with it. It can inspire people to write their own code with ions or get access to one. The question is whether Nature Communications is then the right venue, because here I would have the expectation that there is a useful take-home message for a broad readership.

An interesting additional fact is that in a rather similar paper about ions and SCLC, published this year (ACS Energy Lett. 2020, 5, 376–384), the authors have concluded that they cannot determine the mobility from the SCLC data (and therefore only study upper limits to trap densities).

Reviewer #3 (Remarks to the Author):

The authors addressed precise discussion to the all questions regarding the concerns on the ambiguities in the previous discussion. I do now believe this work with current status is giving good knowledge to the readers about the importance of SCLC measurement on perovskite. I therefore think this manuscript is ready to be published.

REVIEWER COMMENTS (please find our response to the reviewer comments in red)

Reviewer #2 (Remarks to the Author):

Review for the revised version of the paper "Space-charge-limited electron and hole currents in hybrid organic-inorganic perovskites" by Alvar et al.

The key issue I had in the previous round was with the extremely low values of the mobility that do not seem to be consistent with any photovoltaic behaviour. In order to rationalize their findings, the authors have provided a simulated and experimental JV curve using Langevin-type recombination. This implies that they have set the SRH lifetime to infinity, i.e. assume that they are in the radiative limit. Unlike in organics, the bimolecular recombination coefficient found in literature is determined by radiative recombination and can therefore be calculated from the absorption coefficient and intrinsic carrier concentration of the material (see e.g. <https://www.nature.com/articles/s41467-017-02670-2>). Therefore, k_{rad} is known to a certain degree and doesn't depend on mobility. The papers cited by the authors (Sherkar et al. and Wehrenpfennig et al.) also find similar values.

If the typically cited range of mobilities of 1 - 100 cm²/Vs is correct, Langevin type recombination would be much faster than that and it makes no sense to assume that recombination was diffusion limited. Even if due to the presence of grain boundaries the external mobility was much slower (orders of magnitude than that), the Langevin theory should really use the local mobility that corresponds to what is determined by non-contact measurements. So I don't find it credible to use Langevin theory here.

We have provided the preliminary simulation of a solar cell in response to the reviewers' concern that the low mobility would not be consistent with the generally good solar-cell performance of perovskite solar cells. We have indeed used a simple Langevin-type rate here, but we did not intend to state that this is the recombination mechanism, we just wanted to point out that even with a low mobility a solar cell can give good performance, provided that the recombination rate, whatever the mechanism, is low.

We would also like to note that the Langevin rate is based on the sum of the mobilities, and the electron (and even hole mobility) would be reasonable in view of the wide variety in published values (see e.g. Fig. 13 in Peng et al. Chem. Soc. Rev. 46, 5714-5729 (2017), especially when one considers the time scales of the measurements. Whether the time-averaged recombination would only depend on the local (fast time scale) mobility is debatable in our opinion.

The JV curve of the solar cell shows a Voc of ~1.05V for a MAPI cell. Thus, the cell has a Voc that is ~270mV away from the radiative limit (1.32 V for a band gap of 1.6eV). Thus, the non-radiative recombination current exceeds the radiative recombination current by a factor $\exp(270/25.8) = 3.5e4$. Under these circumstances it is most likely not appropriate to calculate the JV curve using what is essentially radiative recombination only. Certainly, part of the voltage loss will be due to interface recombination, part due to bulk recombination. How exactly the loss is distributed, we don't know. But my original argument still holds. The mobilities are too low to explain half way decent photovoltaic functionality in a realistic scenario (of finite SRH lifetimes that everybody reports who looks at transient

optical measurements). The electron mobilities are on the edge of being just about credible if the lifetime of the bulk was very long. But the hole mobilities don't seem to be

credible. What is causing them to be or appear so low, I don't know. Generally, to understand what the authors did in Fig. S7, I would need to see a full list of variables including all contact and interface layers. At the moment, I don't even understand how the curve can reproduce V_{oc} while only having a Langevin recombination rate in the bulk. That should really give the radiative limit if correct values for band gap and effective DOS are used.

One reason why V_{oc} is lower than the radiative limit is because a barrier was used at hole contact, consistent with the lower injection we see when using PTAA in a hole-only device. This lowers the built-in voltage and thus V_{oc} in the model. We do not intend to state that we have only radiative recombination. As stated above, we merely use a simple Langevin-type rate to show that a good efficiency can be obtained even with the mobilities that we find, provided that the recombination rate, whatever the mechanism, is low.

We further would like to emphasize that simulations of solar cells and recombination mechanisms require further extensive investigations (which we will do in the near future) that are outside the scope of this manuscript, which is about the analysis of SCLC in perovskites with mobile ions.

The general problem I have here is that to a certain degree I can only trust the authors that this is what their simulations are giving and there is no other way to simulate the data. As a reviewer (without access to the software and without the time to reproduce everything from scratch) I can only judge whether the data makes sense to me and it still doesn't. What will the publication of the article cause when it is read?

People who do SCLC on perovskites will see that they have to include ions in drift diffusion, which they probably already figured out themselves. Only problem is that 99% cannot do that because they don't have the code for it and no time to program one of their own. So those 99% only have the data as a take home message and that will say what most of you thought the mobility of holes in MAPI is, is off by 6 to 8 orders of magnitude. What will the reader say? He/she will probably just be confused. As I am.

So, the take home message of the paper is not helpful to all those readers, who are not in the author's group and can use the code and I don't see that the paper develops in a direction where it would be helpful. Now one can of course still publish it because it contains interesting insights for the minority of people who can work with it. It can inspire people to write their own code with ions or get access to one. The question is whether Nature Communications is then the right venue, because here I would have the expectation that there is a useful take-home message for a broad readership.

We disagree that our paper is only interesting for a minority of researchers. The main message of the paper is how to interpret SCLC in perovskites. Being a steady-state technique, SCLC analysis is important for understanding the device physics of perovskite devices (operating steady state). Many groups have applied classical SCLC theory to their measurements, which we here show is not applicable. Therefore, this paper is already important to avoid the community being misled by the wrong (classical)

interpretation of SCLC measurements. Importantly, we here show that the frequency and temperature dependence of the permittivity has to be taken into account to analyze SCLC. We demonstrate this by not only modelling the J-V at room temperature, but also the scan rate and temperature dependence, that are all dominated by the permittivity. Other modelling and SCLC papers did not do this, so this is a very important message for the community. For the first time, we have both *measured* and *modelled* the temperature and scan-rate dependence of electron- and hole SCLC currents. Furthermore, we show that SCLC can reveal the sign of the mobile ions based on the direction of the hysteresis. These are the take-home messages of the paper.

While the low hole mobility may be considered surprising, this is not the main take-home message of the paper. However, it demonstrates that steady-state measurements (transport across grain boundaries and potential other factors that may slow down charge transport) give lower mobilities than measurements on short time scales that probe only faster carriers. Previous SCLC analyses of the mobility have been only done by using the classical Mott-Gurney law using the high-frequency permittivity (or otherwise incorrect value), overestimating the mobility and disregarding the other impacts of ions on the J-V characteristics. Our manuscript therefore also demonstrates – besides how SCLC should be interpreted – that steady-state mobilities may be considerably lower than expected.

We would further like to note that such low (hole) mobilities have also been found in field effect transistors. While we acknowledge that the device geometry is different is a FET, and charge-carrier densities are higher, this provides some support that steady-state mobilities can be much lower. We are aware that mobilities in FETs have improved by optimizing the perovskite morphology for FET operation (e.g. single crystals), but this is notoriously difficult and requires a great deal of optimization.

An interesting additional fact is that in a rather similar paper about ions and SCLC, published this year (ACS Energy Lett. 2020, 5, 376–384), the authors have concluded that they cannot determine the mobility from the SCLC data (and therefore only study upper limits to trap densities).

The reason why in the paper by Duijnsteet al. the mobility could not be determined was – at least in part – due to the fact that the authors expected a quadratic classical SCLC regime (as noted by the authors). As we show here, classical SCLC analysis is not applicable to semiconductors with mobile ions. Furthermore, the authors used a low value for the relative permittivity (25.5), whereas we here demonstrate that its frequency and temperature dependence has to be taken into account, giving much higher permittivity values. We believe that the reviewer was set on the wrong foot with the message that the mobility cannot be obtained from SCLC data and other researchers may have concluded the same from this paper. We here demonstrate that it is possible to obtain the mobility from SCLC measurements, by including the effects of ion dynamics and the permittivity.